# The Influence of Bloom Index, Endotoxin Levels and Polyethylene Glycol Succinimidyl Glutarate Crosslinking on the Physicochemical and Biological Properties of Gelatin Biomaterials

**DOI:** 10.3390/biom11071003

**Published:** 2021-07-09

**Authors:** Zhuning Wu, Stefanie H. Korntner, Jos Olijve, Anne Maria Mullen, Dimitios I. Zeugolis

**Affiliations:** 1Regenerative, Modular & Developmental Engineering Laboratory (REMODEL) and Science Foundation Ireland (SFI) Centre for Research in Medical Devices (CÚRAM), National University of Ireland Galway (NUI Galway), H91 TK33 Galway, Ireland; z.wu1@nuigalway.ie (Z.W.); stefanie.korntner@gmail.com (S.H.K.); 2Rousselot BV, 9000 Gent, Belgium; jos.olijve@rousselot.com; 3Teagasc Food Research Centre, Ashtown, D15 DY05 Dublin, Ireland; AnneMaria.mullen@teagasc.ie; 4Regenerative, Modular & Developmental Engineering Laboratory (REMODEL), Università della Svizzera Italiana (USI), 6900 Lugano, Switzerland; 5Regenerative, Modular & Developmental Engineering Laboratory (REMODEL), Charles Institute of Dermatology, Conway Institute of Biomolecular and Biomedical Research and School of Mechanical and Materials Engineering, University College Dublin (UCD), D04 V1W8 Dublin, Ireland

**Keywords:** gelatin biomaterials, bloom index, endotoxin levels, macrophage response

## Abstract

In the medical device sector, bloom index and residual endotoxins should be controlled, as they are crucial regulators of the device’s physicochemical and biological properties. It is also imperative to identify a suitable crosslinking method to increase mechanical integrity, without jeopardising cellular functions of gelatin-based devices. Herein, gelatin preparations with variable bloom index and endotoxin levels were used to fabricate non-crosslinked and polyethylene glycol succinimidyl glutarate crosslinked gelatin scaffolds, the physicochemical and biological properties of which were subsequently assessed. Gelatin preparations with low bloom index resulted in hydrogels with significantly (*p* < 0.05) lower compression stress, elastic modulus and resistance to enzymatic degradation, and significantly higher (*p* < 0.05) free amine content than gelatin preparations with high bloom index. Gelatin preparations with high endotoxin levels resulted in films that induced significantly (*p* < 0.05) higher macrophage clusters than gelatin preparations with low endotoxin level. Our data suggest that the bloom index modulates the physicochemical properties, and the endotoxin content regulates the biological response of gelatin biomaterials. Although polyethylene glycol succinimidyl glutarate crosslinking significantly (*p* < 0.05) increased compression stress, elastic modulus and resistance to enzymatic degradation, and significantly (*p* < 0.05) decreased free amine content, at the concentration used, it did not provide sufficient structural integrity to support cell culture. Therefore, the quest for the optimal gelatin crosslinker continues.

## 1. Introduction

Gelatin is attracting growing attention in the fields of tissue engineering and drug delivery due to its low antigenicity, high cell affinity, relatively easy processability and high availability at low cost [1,2,3,4]. Gelatin is a mixture of peptides produced by partially acid (gelatin type A, isoelectric point of ~8) or alkaline (gelatin type B, isoelectric point of ~5) collagen hydrolysis [5,6]. Subject to the stage of the extraction, gelatin preparations are classified as low (initial stage of extraction/incomplete hydrolysis) or high (late stage of extraction/complete hydrolysis) bloom index (ranging from 50 to 300) [7]. High bloom numbers are associated with high molecular weight gelatin preparations, increased dynamic moduli and increased gelation and melting temperatures [8]. The bloom index and the concentration of the solution affect the gelling capacity and the gel strength of gelatin preparations [9,10], which in turn affect their drug release capacity [11]. Cell response has also been shown to be bloom index-dependent. Indeed, high bloom index gelatin preparations, possibly due to their high viscosity that affects nutrient transport, have been shown to reduce cytocompatibility and to increase inflammatory reaction [12,13].

The use of gelatin in biomedicine is somehow restricted due to its high affinity to endotoxin contamination. Endotoxins are large and complex lipopolysaccharides that are localised at the outer membrane of Gram-negative bacteria [14,15] and are often associated with contamination of medical devices. Specifically, as endotoxins possess a high thermal stability [16] and are hard to destroy with conventional sterilisation conditions [17], residual endotoxins constitute the most significant pyrogen in parenteral drugs and medical devices [18,19] and frequently lead to post-operative complications, such as delayed tissue regeneration and homeostasis, implant failure and septic shock [20,21,22,23]. It is therefore imperative to assess residual endotoxins in medical devices, and especially in natural biomaterials that are prone to endotoxin contamination [24,25].

As gelatin is the product of collagen hydrolysis, exogenous crosslinks should be used to stabilise the produced materials. Unfortunately, inappropriate crosslinking may reduce cytocompatibility and increase inflammation response [26]. To this end, several crosslinking methods have been assessed over the years with contradictive outcomes. For example, although the use of genipin has been advocated as a means to increase mechanical stability and denaturation temperature [27], genipin, similarly to glutaraldehyde, has been shown to induce strong in vivo rejection [28]. Although carbodiimide has been proposed as a gelatin crosslinker as effective as glutaraldehyde [29], the fast degradation rate of the resultant scaffold prohibits its further use [28]. Although transglutaminase has shown promise as a gelatin crosslinker [28,30], its limited crosslinking capacity [31,32] has also restricted its use. Evidently, the quest for the optimal gelatin crosslinking method continues.

Considering the above, herein, we ventured to assess the influence of bloom index, endotoxin levels and 4-arm polyethylene glycol succinimidyl glutarate crosslinking on the physicochemical and biological properties of gelatin-based biomaterials. Specifically, the biophysical, biochemical and biological properties of gelatin (porcine type A and bovine type B, from two different suppliers) and gelatin-based biomaterials (hydrogels for physicochemical analysis and films for biological analysis) were assessed as a function of different levels of endotoxin content (from <1 up to 10,370 endotoxin units per gram), bloom index (from 220 to 355) and crosslinking (non-crosslinked and crosslinked). The 4-arm polyethylene glycol succinimidyl glutarate was selected for crosslinking the gelatin-based biomaterials, as its stabilisation and cytocompatibility efficiency have been repeatedly demonstrated in the literature [33,34,35].

## 2. Materials and Methods

### 2.1. Materials

Gelatin products (Table 1) were provided by Rousselot R&D Center (Ghent, Belgium) or purchased from Sigma-Aldrich (Arklow, Co., Wicklow, Ireland). 4-arm polyethylene glycol (PEG) succinimidyl glutarate (4SG, Mw 10,000) was purchased from JenKem Technology (Plano, TX, USA). All other materials and reagents were purchased from Sigma-Aldrich (Arklow, Co., Wicklow, Ireland) unless otherwise stated.

### 2.2. Electrophoretic Mobility Assessment

To assess the electrophoretic mobility of the gelatin samples, sodium dodecyl sulphate-polyacrylamide gel electrophoresis (SDS-PAGE) under non-reducing conditions was conducted [36] using a Mini-Protean 3 electrophoresis system (Watford, Bio-Rad Laboratories, UK). A 3% running gel and a 5% separation gel were used. Briefly, 10% gelatin solutions were dissolved in 0.5 M acetic acid and neutralised with 1 N NaOH, followed by the addition of 5× sample buffer (bromophenol blue/SDS). The sample-buffer mixtures were heated at 95 °C for 5 min and a 10 µL aliquot of each mixture was loaded into each well of the running gel. High-purity soluble collagen type I (Chaponost, Symatese, France) was used as a control. Electrophoresis was carried out by first applying 50 V constant voltage until the samples reached the end of the running gel (~30 min) and then 120 V constant voltage was applied until the samples reached the end of the separation gel (~60 min). The gels were stained with a silver staining kit (SilverQuest™, Invitrogen, Thermo Fisher Scientific, Waltham, MA, USA) according to the manufacturer’s protocol.

### 2.3. Fabrication and Crosslinking of Gelatin Hydrogels and Films

Gelatin hydrogels (250 μL final volume was used) were prepared by dissolving gelatin products in 1× phosphate buffered saline (PBS) at 50 °C to form solutions of 100 mg/mL, mixing them with 1 mM PEG-4SG and letting them assemble at 25 °C for 1 h in Ace silicone O-rings (Z504165). Gelatin films (250 μL final volume was used) were prepared by dissolving gelatin products in 1× PBS at 50 °C to form a solution of 100 mg/mL, mixing them with 1 mM PEG-4SG at 25 °C in 48-well plates (Sarstedt, Nümbrecht, Germany) and allowing the liquid to evaporate overnight in a fume hood.

### 2.4. Biomechanical Assessment

The mechanical properties of gelatin hydrogels were assessed via uniaxial compression using a universal tensile testing machine (Z2.5, Zwick/Roell, Ulm, Germany), loaded with a 100 N static load cell. Uniaxial constant loading was performed on gelatin hydrogels with approximately 3 mm height and 10 mm diameter. The gelatin hydrogels were placed between two loading cells and compressed until 70% deformation, with a compression rate of 10 mm/min. Force, strain and elastic modulus were determined by plotting stress versus strain curves. Both compression strength and elastic modulus were determined on the linear area of the curves at the position of 30% deformation, and elastic modulus was calculated using the linear equation of trend-lines at the position of 30% of the deformation [37].

Note: The gelatin films (both non-crosslinked and crosslinked) were too fragile in wet state to be mechanically assessed.

### 2.5. Free Amines Assessment

Crosslinking efficiency was quantified using the 2,4,6-trinitrobenzene sulfonic acid (TNBSA) assay (Thermo Fisher Scientific, Dublin, Ireland) [36]. Briefly, gelatin hydrogels (~500 mg) were incubated with TNBSA at 37 °C for 2 h. The reaction was stopped by adding 10% SDS and 1 M hydrochloric acid. The mixtures were subsequently heated at 95 °C for 15 min in order to hydrolyse the gelatin hydrogel samples. The absorbance was read at 335 nm (Varioskan Flash Multimode Reader, Thermo Fisher Scientific, Dublin, Ireland) and values were normalised to the standard curve, which has been generated with a series of known glycine concentrations (0.005, 0.01, 0.02, 0.03, 0.04 and 0.05 mg/mL).

### 2.6. Resistance to Enzymatic Degradation Assessment

Enzymatic stability of the gelatin hydrogels was quantified with the collagenase assay [38]. Briefly, gelatin hydrogels were weighed and then incubated for 2 h in 0.1 M Tris-HCl and 500 mM CaCl_2_ at pH 7.4. Subsequently, the hydrogels were digested with 50 U/mL bacterial collagenase type II (MMP-8; 17101-015, Gibco, Thermo Fisher Scientific, Dublin, Ireland). After 3, 6, 9, 12 and 24 h of digestion at 37 °C, centrifugation was carried out at 10,000× *g* for 5 min, the supernatant was removed and the remaining gelatin hydrogels were weighed. The degree of enzymatic degradation was quantified using the following equation: [(Wo − Wt)/Wo] × 100, where Wo is the original weight and Wt is the remaining weight.

### 2.7. Cell Culture

In vitro inflammatory response was assessed using human-derived leukemic monocyte cells (THP-1, ATCC, Manassas, VA, USA) [39]. Cells were grown in RPMI-1640 medium supplemented with 10% foetal bovine serum and 1% penicillin-streptomycin at 37 °C in a 95% humidified atmosphere of 5% CO_2_. Cells were seeded on gelatin films and tissue culture plastic (TCP) at an initial density of 26,000 cells/cm^2^ and cultured in RPMI-1640 medium for 6 h to enable cell attachment. A mature macrophage-like state was induced through treatment with phorbol 12-myristate 13-acetate (PMA) at 100 ng/mL for 6 h, as has been described previously. Subsequently, human macrophages were washed with Hank’s Balanced Salt Solution (HBSS) and incubated with RPMI-1640 medium at 37 °C in a 95% humidified atmosphere of 5% CO_2_ for 1 and 2 days.

Note: First, we tried to assess the cytocompatibility of the crosslinked gelatin hydrogels using human adipose-derived stem cells. Unfortunately, the 1 mM 4-arm polyethylene glycol succinimidyl glutarate resulted in rapid hydrogel degradation in culture (Appendix A), prohibiting further assessment. For this reason, gelatin films were fabricated and THP-1 response was assessed, as they require only 2 days in culture.

### 2.8. Cell Viability Assessment

Cell (THP-1) viability was analysed using the Live/Dead^®^ assay (Life Technologies, Thermo Fisher Scientific, Dublin, Ireland) as per the manufacturer’s protocol. Briefly, at the end of each timepoint, the gelatin films were washed three times with HBSS and incubated with calcein AM and ethidium homodimer-1 solution (2 µM calcein AM and 4 µM ethdium homodimer-1) in HBSS at 37 °C and a 5% CO_2_ humidified atmosphere for 30 min. The gelatin films were washed with fresh HBSS to remove excess dye. Images were then acquired using an inverted fluorescence microscope (IX 51, Olympus Corporation, Tokyo, Japan). Live (green: FITC, ~495 nm) and dead (red: TexasRed, ~589 nm) cells were analysed using ImageJ software (NIH, Bethesda, MD, USA).

### 2.9. Cell Proliferation Assessment

Cell (THP-1) proliferation was assessed using the Quant-iT^TM^ PicoGreen^®^ dsDNA kit (Invitrogen, Thermo Fisher Scientific, Dublin, Ireland), according to the manufacturer’s guidelines. Gelatin films were washed three times with HBSS at each timepoint, and 200 µL DNase free water was added and frozen at −80 °C until analysis. Gelatin films were freeze-thawed at least three times in order to lyse the cells. Subsequently, PicoGreen^®^ working solution was added to the gelatin films and incubated at room temperature for 5–10 min, protected from light. Fluorescence was measured at excitation and emission wavelengths of 480 and 520 nm respectively, using a Varioskan^TM^ Flash Multimode Reader (Thermo Fisher Scientific, Dublin, Ireland). The obtained values were normalised to the standard curve, which was generated with a series of known DNA stock solutions at different concentrations (0, 5, 10, 25, 50, 100, 500 and 1000 ng/mL).

### 2.10. Cell Metabolic Activity Assessment

Cell (THP-1) metabolic activity was analysed using the alamarBlue^®^ assay (Invitrogen, Thermo Fisher Scientific, Dublin, Ireland), as per the manufacturer’s protocol. Briefly, gelatin films were washed three times with HBSS at the end of each timepoint, and 10% alamarBlue^®^ was added to each gelatin film and incubated at 37 °C in a 5% CO_2_ humidified atmosphere for 3 h. Absorbance was measured at excitation and emission wavelengths of 570 and 600 nm respectively, using a Varioskan^TM^ Flash Multimode Reader (Thermo Fisher Scientific, Dublin, Ireland).

### 2.11. Cell Morphology Assessment

At the end of each timepoint, gelatin films were washed three times with HBSS and cell (THP-1) morphology was analysed using a brightfield microscope (Olympus Corporation, Tokyo, Japan). The levels of macrophage clusters on gelatin films were analysed by measuring cell numbers on different gelatin films using ImageJ software (NIH, Bethesda, MD, USA).

### 2.12. Statistical Analysis

Numerical data are expressed as mean ± standard deviation. Statistical analysis was performed using SPSS (IBM Corporation, Armonk, NY, USA). Analysis was performed using one-way analysis of variance (ANOVA) for multiple comparisons, and two-sample t-tests for pairwise comparisons were employed after confirming the following assumptions: the distribution from which each of the samples was derived was normal (Shapiro–Wilk normality test) and the variances of the population of the samples were equal to one another (Levene’s test for equal variances). Nonparametric statistics were used when either or both of the above assumptions were violated, and consequently, Kruskal–Wallis test for multiple comparisons and Mann–Whitney test for two samples were carried out. Statistical significance was accepted at *p* < 0.05.

## 3. Results

### 3.1. Electrophoretic Mobility Assessment

SDS-PAGE (Figure 1) revealed typical electrophoretic mobility of gelatin for all samples, characterised by the presence of α-, β- and γ- bands, as well as other bands of variable molecular weight.

### 3.2. Biomechanical and Free Amine Assessment

Among the different gelatin preparations, the RAP80 and SBB1.5K resulted in hydrogels with the lowest (*p* < 0.05) stress and modulus values in both non-crosslinked and crosslinked state, the SBB1.5K resulted in hydrogels with the highest % free amines in both non-crosslinked (*p* < 0.05) and crosslinked state (*p* < 0.05), and for all gelatin preparations, crosslinking significantly (*p* < 0.05) increased stress and modulus values and decreased % free amines (Table 2).

### 3.3. Resistance to Enzymatic Degradation Assessment

Collagenase digestion analysis (Figure 2) revealed that all non-crosslinked samples were completely degraded within 6 h of exposure to collagenase. Crosslinking significantly (*p* < 0.05) increased resistance to collagenase digestion, as crosslinked samples had at least >35% remaining mass after 6 h and were completely digested after 12 h of exposure to collagenase. Among the non-crosslinked groups, the SBB1.5K group exhibited the lowest (*p* < 0.05) resistance to collagenase digestion and no significant (*p* > 0.05) differences were observed between the other groups at 3 h of collagenase digestion. Among the crosslinked groups, the SBB1.5K group followed by the RAP80 group exhibited significantly (*p* < 0.05) lower resistance to collagenase digestion than the other groups, and no significant (*p* > 0.05) differences were observed between the other groups at 9 h of collagenase digestion.

### 3.4. Macrophage Viability, Proliferation and Metabolic Activity Assessment

Qualitative (Figure 3A) and quantitative (Figure 3B) cell viability and quantitative DNA concentration (Figure 3C) and metabolic activity (Figure 3D) analyses revealed no significant (*p* > 0.05) differences between the groups at any timepoint, apart from the SBB1.5K group, which exhibited the lowest (*p* < 0.05) cell viability at day 2 and the lowest (*p* < 0.05) metabolic activity at days 1 and 2.

### 3.5. Macrophage Morphology Assessment

Macrophage morphology (Figure 4A) and clusters (Figure 4B) assessment revealed that macrophages were of round morphology in all groups and formed the least (*p* < 0.05) number of clusters on TCP, RAP1, RAP80, RAP780 and RBB9 at both timepoints, respectively.

## 4. Discussion

Gelatin, the hydrolysed derivative of collagen, is extensively used in food, pharma and medical device sectors [40,41,42,43]. In medical device development, bloom index, a unit that measures the extent of hydrolysis, is an important parameter that should be assessed as it affects the gelling capacity, gel strength and cell response of gelatin preparations. Another crucial factor in medical device development is residual endotoxin levels, as endotoxins are considered to be the most significant pyrogens and are associated with post-operative complications, including implant failure. One should also not forget that as gelatin is produced via collagen hydrolysis, exogenous crosslinks should be introduced to enhance mechanical strength, but without compromising cellular functions of gelatin biomaterials. Herein, we assessed the physiochemical and biological properties of gelatin-based biomaterials and we correlated them to their bloom index, amounts of endotoxins present in the original raw materials and polyethylene glycol succinimidyl glutarate crosslinking.

Starting with SDS-PAGE assessment, all gelatin preparations were comprised of α-, β- and γ- bands, as well as other bands of variable molecular weight. This is in agreement with previous publications, considering that gelatin is a mixture of water-soluble protein fragments, obtained by the destruction of collagen, with a molecular weight distribution ranging from 10 to 400 kDa [44]. Variable number and intensity of bands was observed between the different gelatin preparations, which we attribute to the origin and the manufacturing process of the respective gelatin preparations [45,46,47,48,49], as has also been observed previously for collagen preparations [36,39,50,51].

With respect to mechanical properties, resistance to enzymatic degradation and free amine content, crosslinking increased mechanical properties and resistance to enzymatic degradation and decreased free amine content. Within the different gelatin preparations, the RAP80 and SBB1.5K groups exhibited the lowest compression stress and elastic modulus values, the highest free amine content (only the SBB1.5K was statistically significant) and the lowest resistance to enzymatic degradation (i.e., the crosslinked groups at 9 h of degradation). We attribute this low mechanical strength/high free amine content/low resistance to enzymatic degradation to the low bloom strength of these gelatin preparations. In agreement with our observations, previous studies have shown that high bloom rate results in gelatin scaffolds with high ultimate strength, rigidity and Young’s modulus [8,12,52,53]. With respect to free amines and resistance to enzymatic degradation, high % of free amines and low resistance to enzymatic degradation are related to less crosslinked materials, as has been shown repeatedly for gelatin [54,55,56] and collagen [34,36,51] scaffolds.

With respect to cell shape, on all substrates, the cells exhibited a round morphology, whilst cells have been reported to be of round morphology on low bloom index hydrogels and of elongated morphology on high bloom index hydrogels [57,58,59]. We attribute this indifference in cell shape between the different gelatin preparations to residual endotoxins. Indeed, biological analysis with macrophages made it apparent that high levels of residual endotoxins (>1.5 K units/g) were responsible for cell clusters. It has been well-established in the literature that endotoxins are associated with macrophage activation [60,61] and macrophage aggregation is indicative of foreign body response [62,63,64]. Further, high endotoxin levels in gelatin preparations have been shown to significantly increase synthesis of TNF-a and CCL2 [65], notorious pro-inflammatory cytokines [66,67,68]. To reduce the burden of endotoxin-activated immune response, functionalisation strategies have been recommended (e.g., gelatin preparations functionalised with desaminotyrosine or desaminotyrosyl tyrosine resulted in reduced expression of IL6 and TNF-a [69]).

It is interesting to note that gelatin crosslinking has been the subject of many investigations [70,71,72,73], however, similarly to collagen, the ideal crosslinking method still remains elusive. It is also worth noting that several contradictions about a given crosslinking method can be found in the literature (e.g., both positive [27,29] and negative [28] results have been reported for genipin [27] and carbodiimide [29] crosslinking), compromising the utilisation of gelatin in the applied field of medical devices. Considering that polyethylene glycol succinimidyl glutarate has been used extensively as a collagen crosslinker [33,34,35], we hypothesised that it would also effectively crosslink gelatin preparations. Indeed, although the mechanical properties and the resistance to enzymatic degradation were increased and the free amine content was decreased, the produced gelatin hydrogels were substantially degraded after only 7 days in culture with adipose-derived stem cells. A plausible reason could be that we used a very low polyethylene glycol succinimidyl glutarate concentration for gelatin and therefore a dose response should be conducted to conclusively find out the potential of this crosslinking agent in gelatin research.

## 5. Conclusions

In this study, we evaluated the influence of bloom index, endotoxin level and polyethylene glycol succinimidyl glutarate crosslinking on the physicochemical and biological properties of gelatin biomaterials. Our data suggest that low endotoxin and high bloom index gelatin preparations should be used in the medical device sector, unless fast degradation is required, in which case, low endotoxin and low bloom index gelatin preparations should be used. Polyethylene glycol succinimidyl glutarate crosslinking (at 1 mM) was not able to induce notable cell culture stability, imposing the need for the development of alternative crosslinking methods.

## Figures and Tables

**Figure 1 biomolecules-11-01003-f001:**
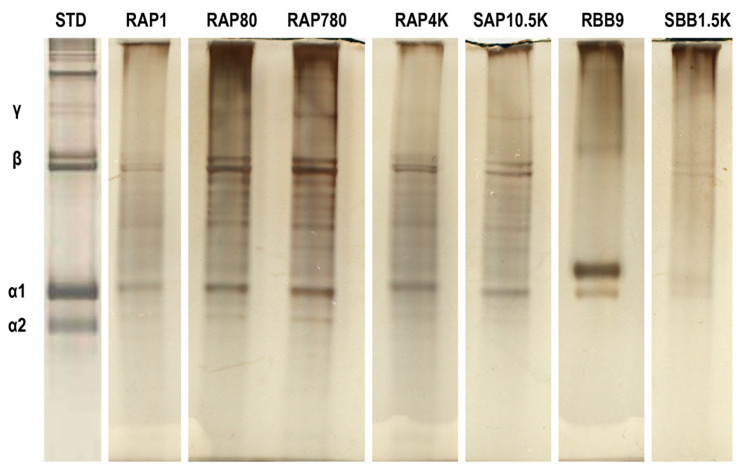
SDS-PAGE analysis of gelatin samples used in this study.

**Figure 2 biomolecules-11-01003-f002:**
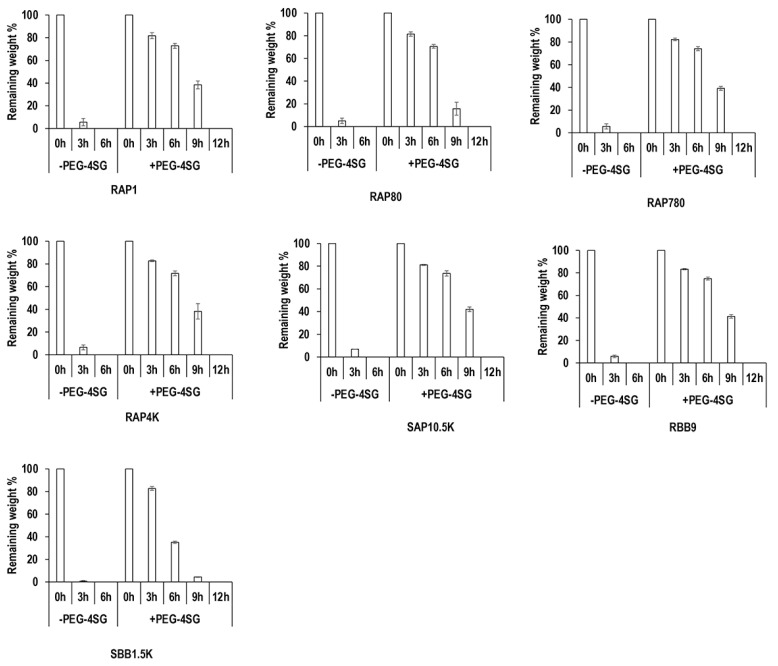
Resistance to enzymatic degradation analysis of non-crosslinked (-PEG-4SG) and crosslinked (+PEG-4SG) gelatin samples used in this study.

**Figure 3 biomolecules-11-01003-f003:**
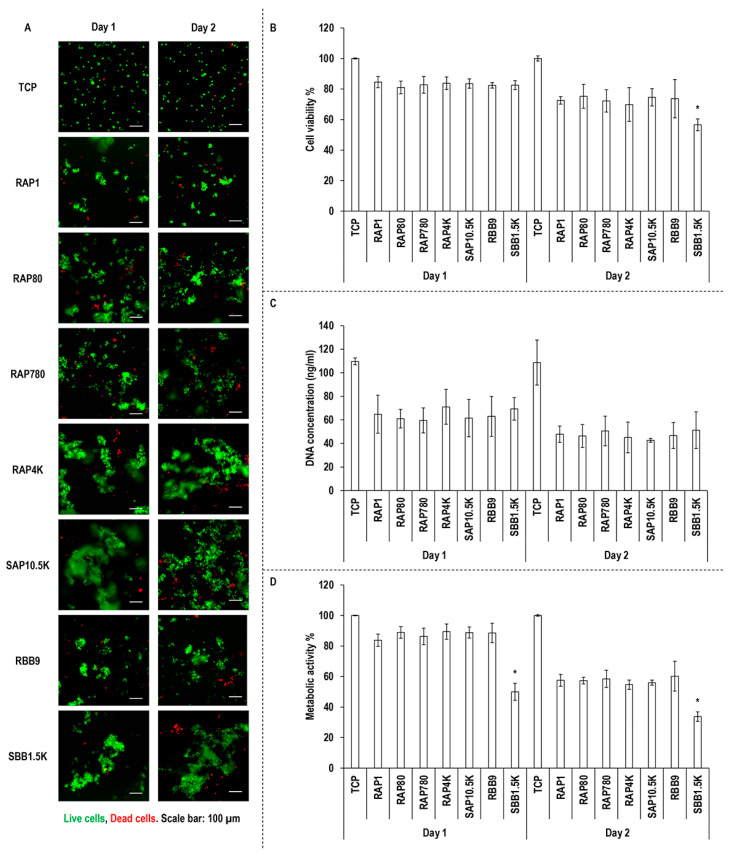
Macrophage viability (**A**,**B**), DNA concentration (**C**) and metabolic activity (**D**) analyses. *: Indicates statistically significant difference (*p* < 0.05) between the gelatin samples used in this study at a given timepoint.

**Figure 4 biomolecules-11-01003-f004:**
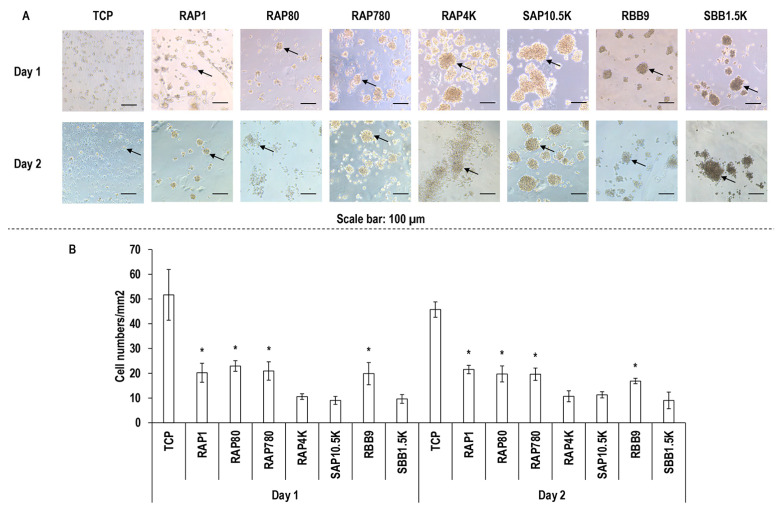
Macrophage morphology (**A**) and cell number (**B**) analyses. Cell clusters are indicated using black arrows. TCP: Tissue culture plastic. *: Indicates statistically significant difference (*p* < 0.005) between the gelatin samples used in this study at a given timepoint.

**Table 1 biomolecules-11-01003-t001:** Properties of gelatin samples used in this study. * The endotoxin units per gram values were provided by the supplier and were measured using the ENDONEXT™ EndoZyme^®^ II–Recombinant Factor C (rFC) Endotoxin Detection Assay (Bernried am Starnberger See, Hyglos GmbH).

Material	Abbreviation	Bloom	Endotoxin Units per gram *
Rousselot, type A porcine	RAP1	355	<1
Rousselot, type A porcine	RAP80	220	80
Rousselot, type A porcine	RAP780	285	780
Rousselot, type A porcine	RAP4K	300	4000
Sigma G2500, type A porcine	SAP10.5K	300	10,370
Rousselot, type B bovine	RBB9	247	9
Sigma G9382, type B bovine	SBB1.5K	225	1360

**Table 2 biomolecules-11-01003-t002:** Mechanical properties and free amine content of non-crosslinked (-PEG-4SG) and crosslinked (+PEG-4SG) gelatin samples used in this study. *: Indicates lowest (*p* < 0.05) mechanical properties and highest (*p* < 0.05) free amine content among non-crosslinked and crosslinked gelatin samples used in this study. #: Indicates higher (*p* < 0.05) mechanical properties and lower (*p* < 0.05) free amine content between non-crosslinked and crosslinked gelatin samples used in this study.

	RAP1		RAP80		RAP780		RAP4K		SAP10.5K		RBB9		SBB1.5K	
	-PEG-4SG	+PEG-4SG	-PEG-4SG	+PEG-4SG	-PEG-4SG	+PEG-4SG	-PEG-4SG	+PEG-4SG	-PEG-4SG	+PEG-4SG	-PEG-4SG	+PEG-4SG	-PEG-4SG	+PEG-4SG
Stress (kPa)	21.7 ± 1.2	28.8 ± 1.6 #	12.5 ± 0.6 *	15.4 ± 0.2 *#	21.0 ± 2.9	28.8 ± 3.2 #	22.1 ± 1.9	27.5 ± 1.8 #	22.8 ± 2.5	29.5 ± 3.0 #	21.8 ± 1.3	30.2 ± 3.1 #	10.2 ± 2.0 *	16.0 ± 2.8 *#
Modulus (kPa)	72.4 ± 4.1	95.5 ± 5.0 #	41.8 ± 1.9 *	51.3 ± 0.4 *#	70.0 ± 9.4	95.2 ± 10.6 #	73.8 ± 6.3	91.6 ± 6.0 #	75.8 ± 8.1	97.8 ± 10.4 #	72.8 ± 4.3	100.6 ± 10.1 #	34.0 ± 6.5 *	53.3 ± 9.2 *#
Free Amines (%)	1.6 ± 0.2	0.6 ± 0.2 #	1.9 ± 0.1	0.8 ± 0.1 #	1.7 ± 0.3	0.6 ± 0.1 #	1.7 ± 0.1	0.7 ± 0.3 #	1.6 ± 0.1	0.8 ± 0.1 #	1.8 ± 0.1	0.8 ± 0.1 #	2.4 ± 0.1 *	1.8 ± 0.1 *#

## Data Availability

Data are available from Dimitrios I. Zeugolis.

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
