# Peer review of "The Influence of Bloom Index, Endotoxin Levels and Polyethylene Glycol Succinimidyl Glutarate Crosslinking on the Physicochemical and Biological Properties of Gelatin Biomaterials"

_biomolecules, 2021, doi:10.3390/biom11071003_

Round 1

Reviewer 1 Report

The present manuscript evaluates the effect of bloom index and endotoxin levels on the physicochemical and biological properties of gelatin biomaterials. The objective seems interesting with potential application in tissue engineering. However, revision is required and some comments should be taken into account:

- Introduction: A more detailed description of the bloom index and gelatin-based biomaterials, including the effect of crosslinking, should be included. Please, the objective of the manuscript must be clearly defined.

- In order to assess the purity, I wonder what the protein concentration is and if it has some other minor component.

- Why have the described conditions been used for the manufacture of gelatin hydrogels and films?

- Why are hydrogels and films made if only the biological properties of the films are then evaluated?

- I think that films should also be mechanically measured by tensile tests and not only the hydrogels. Why are dynamic tests not proposed?

- Please, the table captions should describe the figures and not the results.

- The title refers to the influence of the bloom index but then the data is represented in some measures (for example, mechanical properties) as a function of crosslinking.

Reviewer 2 Report

In this study, the authors investigated how bloom index and endotoxin level of gelatin hydrogel affecting macrophage viability and morphology. Overall, the conclusion “Our data suggest that low endotoxin and high bloom index gelatin preparations should be used in medical device sector” is ambiguous and with absolutely no data to support. First of all, higher or lower compression stress and degradation rate are all relative terms, they need to be put into the context of specific applications to make sense. Also, even though the author showed SBB1.5K groups exhibited the lowest compression stress and lowest cell viability, they attribute it to low bloom index of SBB1.5K, which again, has no data to support, the data did not support RAP1, which has the highest bloom index, has the best cell compatibility and highest compression stress. The experimental design for cell compatibility test also need to improve, more than only one type of cells need to be tested, and two-day observation was too short to reach any meaningful conclusions.

Reviewer 3 Report

In my opinion article entitled Bloom index and endotoxin levels control the physicochemical 2 and biological properties of gelatin biomaterials written by Wu et al. is very interesting and represent high impact on current studies in the field of endotoxin levels control. I am impressed by studies described in this article and I support fully publishing it by Biomolecules journal.

Author Response

Reviewer 3

Comment No 1:In my opinion article entitled Bloom index and endotoxin levels control the physicochemical 2 and biological properties of gelatin biomaterials written by Wu et al. is very interesting and represent high impact on current studies in the field of endotoxin levels control. I am impressed by studies described in this article and I support fully publishing it by Biomolecules journal.

Response: Thank you very much for your positive comments.

Round 2

Reviewer 1 Report

In the revised manuscript, authors made some necessary revision and basically addressed the responses to the questions my comments- even though I consider a better mechanical characterization should be included in order to assess the performance of the gelatin biomaterials. Therefore, I fully accept the revised manuscript

Author Response

We would like to thank the reviewer for the positive feedback.